# Subroutine Embedding and Finite Element Simulation of the Improved Constitutive Equation for Ti6Al4V during High-Speed Machining

**DOI:** 10.3390/ma16093344

**Published:** 2023-04-24

**Authors:** Lijuan Liu, Wenge Wu, Yongjuan Zhao, Yunping Cheng

**Affiliations:** 1School of Mechanical Engineering, North University of China, Taiyuan 030051, China; 2Institute for Civ-Mil Integration & Collaborative Innovation, North University of China, Taiyuan 030051, China

**Keywords:** titanium alloy, high-speed cutting, constitutive equation, subroutine, finite element simulation

## Abstract

The Johnson–Cook (J–C) constitutive model is not suitable for Ti-6Al-4V alloy in the high-speed cutting finite element simulation, as it has no response dynamic recrystallization softening effect under heavy impact and high temperature. In this paper, an improved constitutive model considering the recrystallization effect was established, and the parameters were fitted with the data of flow stress–strain of the Split Hopkinson Pressure Bar (SHPB) test. The relevant theories of cutting finite element simulation were studied, such as nonlinear constitutive elastic–plastic deformation, strain state, and material yield. A subroutine that included the Recht shear failure instability criterion and the improved model was coded in Fortran and embedded in the finite element simulation software AdvantEdge FEM, along with the return mapping stress integration algorithm. The simulated stress of the improved model dropped dramatically from 460 MPa to 220 MPa when the temperature rises from 950 °C to 1000 °C, and its decline reached 46.7%, while the J–C model only decreased by 10%. Comparative studies indicate that the stress change of the improved constitutive simulation is closer to the SHPB test results than the J–C constitutive, and the new one is more suitable when it expresses the high temperature and heavy impact in the high-speed milling.

## 1. Introduction

Titanium alloy is a widespread application in all fields of life, such as marine, automotive, aviation, sports, medicine, and so on because of its good plasticity, high strength, temperature resistance, and corrosion resistance [1,2,3]. It is true that the titanium alloys have a relatively low level of processing efficiency and quality, and the recommended cutting speed is 30~50 m/min with cemented carbide tools in production [4,5]. In general, the production process will be made difficult if the cutting speed exceeds 30 m/min and 60 m/min with a high-speed steel tool or a cemented carbide tool, respectively [6]. Cutting is considered high speed when the cutting speed of titanium alloy reaches 100 m/min [7,8,9,10].

It develops normally believed that the mechanical properties of materials at high deformation rates are different to those under quashing static loads. The flow stress of most metals grows with the increase of strain rates and are accompanied by strain-rate hardening effect, and this also depends on the ambient temperature time [11]. The constitutive models of different materials are different. Even for the same material, the constitutive models are not the same due to different treatment processes [12]. Therefore, determining the mechanical behavior of materials under heavy temperature, high strain rate, and large strain conditions is of great significance for cutting simulation, as is establishing a constitutive model under these conditions.

Scholars in various countries have studied how to establish a constitutive relationship suitable for the cutting process, and to obtain accurate constitutive relationship parameters and characterize the stress–strain relationship in the high-speed cutting process. In recent years, researchers from various countries have developed various cutting deformation constitutive equations using various experimental methods, in order to describe the cutting deformation characteristics of metal materials as accurately as possible. Typical metal cutting constitutive equations include the Johnson–Cook constitutive model, the Baummann–Chiesa–Johnson (BCJ) model [13], the Marusich constitutive model [14], and the Nemat–Nasser model [15]. A dislocation mechanical model called the Zerilli–Armstrong model reflects the effects of the solute and grain size on the constitutive model from material microstructure [16]; Rhim and Oh put forward the constitutive model of AISI1045 steel for high speed cutting [17]; Calamaz [18] and Özel et al. [19] studied the strain softening phenomenon of constitutive models, and applied the constitutive model to the research of chip and tool wear; Cheng Guoqiang et al. added a softening term in the constitutive model to reflect the damage to the materials [20]; Peng Jianxiang et al. studied the stress and strain of tantalum at different temperatures and strain rates, and proposed an improved Zerilli–Armstrong model [21]; Li Haitao et al. combined Tanaka’s phase transition theory to propose a two-phase hybrid constitutive model for shape memory alloys [22]; Jun Zhang et al. considered the effect of the strain rate on the initial yield stress and strain hardening, established an improved Johnson–Cook constitutive model, and found that the numerical calculation curve was in good agreement with the test curve [23]. Wang Hongjian et al. finished a finite element simulation of milling force with the Johnson–Cook constitutive model using Deform-3D. The results show that a simulation can better reflect the change of the milling force in the actual milling process [24].

The existing constitutive models are rarelydeveloped to allow for the special properties of a material, and are unable to accurately represent the high-speed cutting characteristics of materials. They do not take into account the influence of the recrystallization softening effect on the constitutive material, especially with regard to the strain softening phenomenon when the strain is greater than the critical strain, while focusing on the parameter modification of the existing constitutive model, and cannot accurately express various phenomena in the cutting process. In this paper, an improved J–C constitutive model of Ti-6Al-4V alloy based on the recrystallization softening effect is established by the incremental method from the bottom layer of the finite element simulation software AdvantEdge FEM 5.5. A subroutine is coded and embedded in the software with the return mapping stress integration algorithm, and the Recht shear instability model is used. The finite element results show that the improved J–C constitutive model is more suitable than the J–C intrinsic model in expressing the stress–strain changes in high-speed milling, making up for the shortcomings of the J–C constitutive model in representing the finite element simulation of high-speed milling, and providing a theoretical basis for the manufacturing process of titanium alloy Ti-6Al-4V.

## 2. Material Constitutive Equation Based on Recrystallization of Ti6Al4V

### 2.1. Test Materials

Table 1 displays the composition of titanium alloy Ti6Al4V, and its physical properties are shown in Table 2.

### 2.2. Test Scheme

The equipment and scheme of the Hopkinson compression bar test and high-speed milling test are shown in Table 3.

### 2.3. The True Flow Stress–Strain Relation

The stress–strain relation curves at different temperatures and strain rates fitted by the SHPB experiment are shown in Figure 1 and Figure 2.

### 2.4. The Improved Material Constitutive Model

The Johnson–Cook (J–C) model is the constitutive equation of metal cutting deformation, which reflects the deformation of metal materials under great strain and the strain-rate at elevated temperature conditions [25]. Equation (1) is the J–C constitutive model that has three terms, strain hardening, strain rate hardening, temperature softening:(1)$$\sigma =(A+B\varepsilon^{n})[1+C(ln(1+\frac{\dot{\varepsilon }}{{\dot{\varepsilon }}_{0}}))][1-{\left(\frac{T-T_{r}}{T_{m}-T_{r}}\right)}^{m}]$$

$\dot{\varepsilon }$: equivalent plastic strain rate; ${\dot{\varepsilon }}_{0}$: reference plastic strain rate; $T_{r}$, $T_{m}$: room temperature and the melting point of the material; *A*: original yield strength at room temperature; *B*: strain hardening coefficient; *C*: strain rate sensitivity; *m*: thermal softening effect; *n*: strain hardening effect.

Ti-6Al-4V undergoes great deformation while high speed milling. The dislocation rearrangement occurs, which prevents the resistance of local plastic deformation from decreasing, and causes stress softening when the temperature is near the recrystallization temperature [26]. There is a critical strain value of 0.25 [27] with the growth of the strain. It exhibits the strain hardening phenomenon when ε <0.25, and the softening effect is serious when ε ≥ 0.25. It is more reasonable that the constitutive model has different expressions to describe the stress changes of all strain intervals. The improved J–C constitutive model is shown in Equation (2).
(2)$$\left\{\begin{array}{l} \sigma =(A+B\varepsilon^{n})[1+C(ln(1+\frac{\dot{\varepsilon }}{{\dot{\varepsilon }}_{0}}))][1-{\left(\frac{T-T_{r}}{T_{m}-T_{r}}\right)}^{m}]\frac{1}{1-[1-\frac{{\left(\sigma_{f}\right)}_{bc}}{{\left(\sigma_{f}\right)}_{ac}}]Fix(\frac{T}{T_{c}})}\;\;\;\;\;\;\;\;\;\;\;\;\;\;\;\;\;\;\;\;\;\;\;\;\;\;\;\;\;\;\;\;\;\;\;\;\;\;\;\;\;\;\;\;\;\;\;\;\;\;\;\;\;\;\;\varepsilon <0.25\;\;\;\;(a)\;\;\;\;\\ \sigma =(A+B\varepsilon^{n}\frac{1}{\exp(\varepsilon^{t})})[1+C(ln(1+\frac{\dot{\varepsilon }}{{\dot{\varepsilon }}_{0}}))][1-{\left(\frac{T-T_{r}}{T_{m}-T_{r}}\right)}^{m}]\left\{1-\left[{\left(\frac{T}{T_{m}}\right)}^{r}\{1-\frac{{\left(\sigma_{f}\right)}_{ac}}{{\left(\sigma_{f}\right)}_{bc}}/{\left[\tanh\varepsilon \right]}^{s}\}\right]\right\}\;\;\;\;\;\;\;\;\varepsilon \geq 0.25\;\;\;\;(b) \end{array}\right.\;\;\;\;\;\;\;\;\;\;$$
$\dot{\varepsilon }$, ${\dot{\varepsilon }}_{0}$, $T_{r}$, $T_{m}$ have the same meaning as Equation (1). $T_{c}$: recrystallization temperature; ${\left(\sigma_{f}\right)}_{bc}$, ${\left(\sigma_{f}\right)}_{ac}$: flow stress before and after recrystallization; *r*, *s*, *t*:*r* = 1, *s* = 0.05, *t* = 2 (for Ti-6Al-4V).

At the beginning of the process, the deformation and the temperature are not high, and the J–C model can well reflect the deformation process of the material. However, while the cutting temperature gradually increases in line with the phase transition temperature and the strain is less than the critical strain value, the stress ratio before and after recrystallization is added to the constitutive by the characteristics of the integral function, which reflects the softening effect of dynamic recrystallization. The larger the deformation, the more severe the strain, until the critical strain value is exceeded, the constitutive shows the characteristics of strong softening, and the softening degree is reflected by the parameters *t* and *s* in the constitutive. The improved model is shown in Equation (3):(3)$$\left\{\begin{array}{l} \sigma =(923.2+673.54\varepsilon^{0.466})\cdot \left[1+0.0167\ln\left(1+\frac{\dot{\varepsilon }}{{\dot{\varepsilon }}_{0}}\right)\right]\cdot \left[1-{\left(\frac{T-T_{r}}{T_{m}-T_{r}}\right)}^{0.73}\right]\cdot {\left[1+0.47\mathrm{fix}(\frac{T}{T_{c}})\right]}^{-1}\;\;\;\;\;\;\;\;\;\;\;\;\;\;\;\;\;\;\;\;\;\;\;\;\;\;\;\;\;\;\;\;\;\;\;\;\;\;\;\;\;\;\;\;\;\;\;\;\;\;\;\;\;\;\;\;\;\;\varepsilon <0.25\;\;\;\;(a)\;\;\;\;\\ \sigma =\left(923.2+673.54\varepsilon^{0.466}\frac{1}{\exp(\varepsilon^{2})}\right)\cdot \left[1+0.0167\ln\left(1+\frac{\dot{\varepsilon }}{{\dot{\varepsilon }}_{0}}\right)\right]\cdot \left[1-{\left(\frac{T-T_{r}}{T_{m}-T_{r}}\right)}^{0.73}\right]\cdot \left\{1-\left[\left(\frac{T}{T_{m}})\cdot \{1-\frac{{\left(\sigma_{f}\right)}_{ac}}{{\left(\sigma_{f}\right)}_{bc}}/{\left[\tanh\varepsilon \right]}^{0.05}\right\}\right]\right\}\;\;\;\;\;\;\;\;\;\varepsilon \geq 0.25\;\;\;\;(b) \end{array}\right.$$

## 3. Subroutines Based on AdvantEdge FEM

### 3.1. Recht Shear Instability Model

The shear instability phenomenon occurs while the slope of the stress–strain is zero according to the Recht criterion, and the abrupt shearing emerges in the plastic deformation zone inside the material. That is, shear instability occurs when the material’s strain hardening rate is balanced with the softening effect. This shows that the strain hardening slope is less than or equal to zero with the reduced shear stress and elevated shear strain, as shown in Equation (4)
(4)$$\frac{d\bar{\tau }}{d\bar{\gamma }}\leq 0\;\;\;\;\;\;\;\;\frac{d\bar{\tau }}{d\bar{\gamma }}=\frac{\partial \bar{\tau }}{\partial \bar{\gamma }}+\frac{\partial \bar{\tau }}{\partial T}\frac{dT}{d\bar{\gamma }}$$

The instability model can be represented by R, as (5):(5)$$R=\frac{\frac{\partial \bar{\tau }}{\partial \bar{\gamma }}}{\frac{-\partial \bar{\tau }}{\partial T}\frac{dT}{d\bar{\gamma }}}$$

$\bar{\tau }$: shear stress; $\bar{\gamma }$: shear strain; *T:* temperature.

The range of *R* is [0, 1]. When *R* equals zero, thermoplastic instability begins to occur; $\bar{\tau }$ and $\bar{\gamma }$ can be expressed in terms of $\bar{\tau }=\frac{\bar{\sigma }}{\sqrt{3}}$ and $\bar{\gamma }=\bar{\varepsilon }\sqrt{3}$. The Recht instability model of the J–C model can be solved by substituting the J–C constitutive model, as shown in Equation (6):(6)$$\bar{\tau }=\frac{1}{\sqrt{3}}\left[A+B{\left(\frac{\bar{\gamma }}{\sqrt{3}}\right)}^{n}\right]\left[1+Cln(1+\frac{\dot{\bar{\gamma }}}{{\dot{\bar{\gamma }}}_{0}})\right]\left[1-{\left(\frac{T-T_{r}}{T_{m}-T_{r}}\right)}^{m}\right]$$

The partial differential of the shear strain is solved by Equation (6), and the strain hardening $\frac{\partial \bar{\tau }}{\partial \bar{\gamma }}\;$ is obtained:(7)$$\frac{\partial \bar{\tau }}{\partial \bar{\gamma }}=\left[\frac{nB}{3}{\left(\frac{\bar{\gamma }}{\sqrt{3}}\right)}^{n-1}\right]\left[1+C\ln\left(1+\left(\frac{\dot{\bar{\gamma }}}{{\dot{\bar{\gamma }}}_{0}}\right)\right)\right]\left[1-{\left(\frac{T-T_{r}}{T_{m}-T_{r}}\right)}^{m}\right]$$

The partial differential of Equation (6) for temperature *T* gets thermal softening $\frac{\partial \bar{\tau }}{\partial T}\;$:(8)$$\frac{\partial \bar{\tau }}{\partial T}=\frac{1}{\sqrt{3}}\left[A+B{\left(\frac{\bar{\gamma }}{\sqrt{3}}\right)}^{n}\right]\left[1+C\ln\left(1+\frac{\dot{\bar{\gamma }}}{{\dot{\bar{\gamma }}}_{0}}\right)\right]\left[\frac{-m}{T-T_{r}}{\left(\frac{T-T_{r}}{T_{m}-T_{r}}\right)}^{m}\right]$$

According to the Recht model, the heating rate per unit area A is determined by $q=\frac{\tau L}{W}\dot{\gamma }$ (*q*: the heat generation rate per unit area; τ: the shear strength of the weak zone; *L*: the length of specimen; $\dot{\gamma }$: the average shear strain rate, equivalent to $\frac{\dot{x}}{L}$; *W*: equivalent to the heat generated during work). According to Carslaw and Jaeger ‘s explanation of temperature, the instantaneous temperature *T_A_* per unit area A on a constant heated infinite medium plane is shown in Formula (9):(9)$$T_{A}=\frac{\tau_{y}L\dot{\gamma }}{W}\sqrt{\frac{t}{\pi \kappa \rho C}}$$

$\tau_{y}$: initial shear yield strength; *k*: thermal conductivity; ρ: specific gravity; *C*: specific heat; *t*: time.

The differential equation of Equation (9) relative to time t is as follows:(10)$$dT_{A}=\frac{\tau_{y}L\dot{\gamma }}{2W}\sqrt{\frac{1}{\pi \kappa \rho Ct}}dt$$
(11)$$\frac{dT_{A}}{d\gamma }=\frac{\tau_{y}L}{2W}\sqrt{\frac{\dot{\gamma }}{\pi \kappa \rho C(\gamma -\gamma_{y})}}$$

The Recht model of the J–C constitutive is obtained by Formulas (5), (7), (8), and (11), as shown in Equation (12):(12)$$R=\frac{\left[\frac{nB}{3}{\left(\frac{\bar{\gamma }}{\sqrt{3}}\right)}^{n-1}\right]\left[1-{\left(\frac{T-T_{r}}{T_{m}-T_{r}}\right)}^{m}\right]}{\frac{1}{\sqrt{3}}\left[A+B{\left(\frac{\bar{\gamma }}{\sqrt{3}}\right)}^{n}\right]\left[\frac{m}{T-T_{r}}{\left(\frac{T-T_{r}}{T_{m}-T_{r}}\right)}^{m}\right]\left[\frac{1}{2}\frac{\tau_{y}L}{W}\sqrt{\frac{\dot{\gamma }}{\pi \kappa \rho C\left(\gamma -\gamma_{y}\right)}}\right]}$$

In the same way, the Recht model of the improved J–C constitutive can be calculated. The J–C model is directly used to reflect the stress–strain relationship of Ti-6Al-4V when the temperature does not reach the recrystallization point and ε<0.25 . While it is greater than the recrystallization temperature, the improved constitutive adds a coefficient of 0.68 to the J–C model, which increases the softening degree of the material. In conclusion, the shear instability model directly uses the Recht model of the J–C constitutive while ε<0.25 , but when the temperature is greater than the recrystallization temperature, a coefficient of 0.68 is added. The improved model is obviously different from the J–C constitutive, and it must be recalculated when ε≥0.25.

Let $U=A+B\cdot {\left(\frac{\bar{\gamma }}{\sqrt{3}}\right)}^{n}\cdot \frac{1}{\exp\left({\left(\frac{\bar{\gamma }}{\sqrt{3}}\right)}^{t}\right)}$, $V=1+C\cdot \ln\left(1+\frac{\overset{•}{\varepsilon }}{\overset{•}{\varepsilon_{o}}}\right)$,
$Y=1-{\left(\frac{T-T_{r}}{T_{m}-T_{r}}\right)}^{m}$, X=1−TTmr·1−σfacσfbctanhγ¯3s, then:(13)$$\frac{\partial U}{\partial \bar{\gamma }}=\frac{B\cdot {\bar{\gamma }}^{n-1}}{{\left(\sqrt{3}\right)}^{n}\exp\left({\left(\bar{\gamma }/\sqrt{3}\right)}^{t}\right)}\cdot \left[n-t{\left(\frac{\bar{\gamma }}{\sqrt{3}}\right)}^{t}\right]$$
(14)∂X∂γ¯=s3·TTmr·σfacσfbctanh(γ3)s+1·1ch2(γ3)
(15)$$\frac{\partial Y}{\partial T}=-m\cdot \frac{{\left(T-T_{r}\right)}^{m-1}}{{\left(T_{m}-T_{r}\right)}^{m}}$$
(16)∂X∂T=−rTmr·1−σfacσfbctanhγ3s·Tr−1
(17)$$\frac{dT}{d\gamma }=\frac{\tau_{y}L}{2W}\sqrt{\frac{\dot{\gamma }}{\pi \kappa \rho c\left(\gamma -\gamma_{0}\right)}}$$

Substituting Equations (13)–(17) into Equation (5), the Recht model of the improved constitutive is obtained as shown in Equation (18):(18)$$R=-\frac{\frac{\partial U}{\partial \bar{\gamma }}\cdot Y\cdot X+U\cdot Y\cdot \frac{\partial X}{\partial \bar{\gamma }}}{\left(U\cdot \frac{\partial Y}{\partial T}\cdot X+U\cdot Y\cdot \frac{\partial X}{\partial T}\right)\cdot \frac{dT}{d\bar{\gamma }}}$$

### 3.2. Subroutine Development

#### 3.2.1. Elastic–Plastic Deformation of Material Nonlinear Constitutive

(1)Stress state

The principal stress $\sigma_{N}$ may be expressed as $\sigma_{N}{}^{3}-I_{1}\sigma_{N}{}^{2}-I_{2}\sigma_{N}-I_{3}=0$, where *I_1_* is the first invariant of the stress tensor, *I_2_* is the second invariant, *I_3_* is the third invariant. Their values are independent of the orientation of the coordinate axis, as shown in Equation (19):(19)$$\left\{\begin{array}{l} I_{1}=\sigma_{x}+\sigma_{y}+\sigma_{z}\\ I_{2}=-(\sigma_{x}\sigma_{y}+\sigma_{y}\sigma_{z}+\sigma_{z}\sigma_{x})+(\tau_{xy}{}^{2}+\tau_{yz}{}^{2}+\tau_{zx}{}^{2})\\ I_{3}=\sigma_{x}\sigma_{y}\sigma_{z}+2\tau_{xy}\tau_{yz}\tau_{zx}-\sigma_{x}\tau_{yz}{}^{2}-\sigma_{y}\tau_{zx}{}^{2}-\sigma_{z}\tau_{xy}{}^{2} \end{array}\right.$$

The stress tensor is decomposed into two parts:$$\left[\begin{array}{l} \sigma_{x}\;\;\tau_{xy}\;\;\tau_{zx}\\ \tau_{xy}\;\;\sigma_{y}\;\;\tau_{yz}\\ \tau_{zx}\;\;\tau_{yz}\;\;\sigma_{z} \end{array}\right]=\left[\begin{array}{l} \sigma_{m}\;\;0\;\;\;\;0\\ 0\;\;\;\;\sigma_{m}\;\;0\\ 0\;\;\;\;0\;\;\;\;\sigma_{m} \end{array}\right]+\left[\begin{array}{l} \sigma_{x}-\sigma_{m}\;\;\;\;\tau_{xy}\;\;\;\;\;\;\;\tau_{zx}\\ \tau_{xy}\;\;\;\;\sigma_{y}-\sigma_{m}\;\;\;\;\tau_{yz}\\ \tau_{zx}\;\;\;\;\;\;\;\;\tau_{yz}\;\;\;\;\sigma_{z}-\sigma_{m} \end{array}\right]$$
$$\left[\begin{array}{l} S_{x}\;\;\;\;S_{xy}\;\;\;S_{zx}\\ S_{xy}\;\;\;S_{y}\;\;\;\;S_{yz}\\ S_{zx}\;\;\;S_{yz}\;\;\;S_{z} \end{array}\right]=\left[\begin{array}{l} \sigma_{x}-\sigma_{m}\;\;\;\;\tau_{xy}\;\;\;\;\;\;\;\tau_{zx}\\ \tau_{xy}\;\;\;\;\sigma_{y}-\sigma_{m}\;\;\;\;\tau_{yz}\\ \tau_{zx}\;\;\;\;\;\;\;\;\tau_{yz}\;\;\;\;\sigma_{z}-\sigma_{m} \end{array}\right],\;\delta_{ij}=\left[\begin{array}{l} 1\;\;0\;\;0\\ 0\;\;1\;\;0\\ 0\;\;0\;\;1 \end{array}\right]$$
The stress tensor can be written as:(20)$$\sigma_{ij}=\sigma_{m}\delta_{ij}+S_{ij}$$

The first tensor on the right side of Equation (20) is the spherical stress and the second is the deviatoric stress. The spherical stress causes the change of elastic volume, and the deviatoric stress leads to the variation of material shape, but no change of volume. Similarly, the deviatoric stress is a stress state, and has a principal direction and invariant, which can be expressed as:(21)$$J_{1}=S_{x}+S_{y}+S_{z}=S_{ii}=0$$
(22)$$\begin{array}{l} J_{2}=-(S_{x}S_{y}+S_{y}S_{z}+S_{z}S_{x})+(S_{xy}{}^{2}+S_{yz}{}^{2}+S_{zx}{}^{2})\\ \;\;\;\;=\frac{1}{2}S_{ij}S_{ij} \end{array}$$
(23)$$\begin{array}{l} J_{3}=S_{x}S_{y}S_{z}+2S_{xy}S_{yz}S_{zx}-S_{x}S_{yz}{}^{2}-S_{y}S_{zx}{}^{2}-S_{z}S_{xy}{}^{2}\\ \;\;\;\;=\frac{1}{3}S_{ij}S_{jk}S_{ki} \end{array}$$

The equivalent stress can be expressed with the stress deviator, as in Equation (24), or with $J_{2}$, as in Equation (25). It can be seen that the equivalent stress is not related to the spherical stress, but is only related to the deviatoric stress.
(24)$$\begin{array}{l} \sigma_{i}=\sqrt{\frac{3}{2}}\sqrt{S_{x}{}^{2}+S_{y}{}^{2}+S_{z}{}^{2}+2(S_{xy}{}^{2}+S_{yz}{}^{2}+S_{zx}{}^{2})}\\ \;\;\;\;=\sqrt{\frac{3}{2}}\sqrt{S_{ij}S_{ij}} \end{array}$$
(25)$$\sigma_{i}=\sqrt{3J_{2}}$$

(2)Strain state

Decompose the strain tensor into a strain sphere tensor that only causes volumetric strain and a strain offset that changes the shape of the material, such as in Equation (26):(26)$$\varepsilon_{ij}=\varepsilon_{m}\delta_{ij}+e_{ij}\;\;$$
where the strain deflection is represented by the following formula:(27)$$e_{ij}=\left[\begin{array}{l} e_{xx}\;\;e_{xy}\;\;e_{zx}\\ e_{xy}\;\;e_{yy}\;\;e_{yz}\\ e_{zx}\;\;e_{yz}\;\;e_{zz} \end{array}\right]=\left[\begin{array}{l} \varepsilon_{x}-\varepsilon_{m}\;\;\;\;\;\;\varepsilon_{xy}\;\;\;\;\;\;\varepsilon_{zx}\\ \varepsilon_{xy}\;\;\;\;\;\;\varepsilon_{y}-\varepsilon_{m}\;\;\;\;\;\;\varepsilon_{yz}\\ \varepsilon_{yz}\;\;\;\;\;\;\;\;\;\varepsilon_{yz}\;\;\;\;\;\;\varepsilon_{z}-\varepsilon_{m} \end{array}\right]$$

The equivalent strain can be determined by Equation (27):$$\begin{array}{l} \varepsilon_{i}=\frac{1}{\sqrt{2}(1+\nu )}\sqrt{{\left(\varepsilon_{x}-\varepsilon_{y}\right)}^{2}+{\left(\varepsilon_{y}-\varepsilon_{z}\right)}^{2}+{\left(\varepsilon_{z}-\varepsilon_{x}\right)}^{2}+\frac{3}{2}(\gamma_{xy}{}^{2}+\gamma_{yz}{}^{2}+\gamma_{zx}{}^{2})}\\ \;\;\;\;=\frac{1}{\sqrt{2}(1+\nu )}\sqrt{{\left(\varepsilon_{1}-\varepsilon_{2}\right)}^{2}+{\left(\varepsilon_{2}-\varepsilon_{3}\right)}^{2}+{\left(\varepsilon_{3}-\varepsilon_{1}\right)}^{2}} \end{array}$$

When the Poisson’s ratio is close to 0.5, the equivalent strain can be expressed by the strain deflection as: $\varepsilon_{i}^{p}=\sqrt{\frac{2}{3}}\sqrt{e_{ij}e_{ij}}$; the equivalent shear strain can be expressed as: $\gamma_{i}=2\sqrt{J_{2}}=\sqrt{3}\varepsilon_{i}$. When the material enters the plastic state, the strain increment at a point can be decomposed as $d\varepsilon_{ij}=d\varepsilon_{ij}^{e}+d\varepsilon_{ij}^{p}$.

The elastic strain increment meets Hooke’s law $d\varepsilon_{ij}^{e}=\frac{1}{2G}d\sigma_{ij}-\frac{3\nu }{E}d\sigma_{m}\delta_{ij}$. The result of Drucker’s formula derivation is $d\varepsilon_{ij}^{p}=d\lambda \frac{\partial f}{\partial \sigma_{ij}}$. The constitutive relation in an incremental form can be obtained as shown in Equation (28):(28)$$d\varepsilon_{ij}=\frac{1}{2G}d\sigma_{ij}-\frac{3\nu }{E}d\sigma_{m}\delta_{ij}+d\lambda \frac{\partial f}{\partial \sigma_{ij}}$$

(3)Yield of material

Materials follow the Huber–Mises yield criterion $f(\sigma_{ij})=0$. It can also be written as: $f(\sigma_{ij})=\sigma_{i}-Y(\lambda )=0$, from Formula (24), we can get:(29)$$f(\sigma_{ij})=\sqrt{\frac{3}{2}S_{ij}S_{ij}}-Y(\lambda )$$

According to the Mises flow criterion:(30)$$d\varepsilon_{ij}^{p}=d\lambda \cdot S_{ij}$$

From Equation (30) and plastic incompressibility ($d\varepsilon_{ii}^{p}=0$) $d\varepsilon_{ij}^{p}=de_{{}_{ij}}^{p}$, that is $de_{ij}^{p}=d\lambda \cdot S_{ij}$
(31)$$\frac{d\varepsilon_{ij}^{p}}{dt}=\frac{d\lambda }{dt}\frac{\partial f}{\partial \sigma_{ij}}=\frac{d\lambda }{dt}\vec{n}$$

$\vec{n}=\frac{3}{2}\frac{S_{ij}}{\sqrt{\frac{3}{2}S_{ij}S_{ij}}}$ represents the direction of the yield trajectory, $\frac{d\lambda }{dt}$ is a scalar: it represents the equivalent plastic strain rate. Similarly, there are the following expressions $de_{}^{p}=\frac{d\varepsilon_{ij}^{p}}{dt}=\frac{d\lambda }{dt}\vec{n}$.

The material yields when the second invariant of stress reaches the yield stress $\sigma^{0}$ as the material deforms. The plastic deformation after yielding is $de_{}^{p}=d{\bar{e}}^{p}\vec{n}$. This equation is solved by a backward method to obtain the following equation: $\Delta e^{p}=\Delta {\bar{e}}^{p}\vec{n}$. In addition, the material partial strain increment $\Delta e=\Delta e^{p}+\Delta e^{e}$, gets $\Delta e^{e}=\Delta e-\Delta e^{p}\vec{n}$, substitute into Equation (31), to get Equation (32):(32)$$(1+\frac{3G}{\sqrt{\frac{3}{2}S_{ij}S_{ij}}}\Delta {\bar{e}}^{p})S_{ij}=2G(e^{e}\left|{}_{t}+\Delta e)\right.$$

Find the inner product of the left and right sides of Equation (32), combined with Newton’s iterative method $x(n+1)=x(n)-\frac{f(x)}{f\prime (x)}$. Let *H* = $\frac{d\bar{\sigma }}{d{\bar{e}}^{p}}$, $\hat{e}=e^{e}\left|{}_{t}\right.+\Delta e$, and the correction of each iteration is x(n−1)−x(n), the increment is calculated to be:(33)$$\delta \lambda^{\left(k\right)}=\frac{3G(\sqrt{\frac{2}{3}\hat{e}:\hat{e}}-\Delta {\bar{e}}^{p})-\bar{\sigma }}{3G+H}$$

The whole process of stress updating in the incremental step can be obtained by Equation (33) and the stress updating algorithm.

#### 3.2.2. Stress Update Algorithm

AdvantEdge FEM provides a mat_user’s interface subroutine. When the constitutive model is imported into AdvantEdge FEM, the user subroutine is compiled into a dynamic link library and loaded dynamically. The subroutine calculates the stress change of the material and returns the Cauchy stress to the AdvantEdge FEM, while the engine passes the deformation rate D and the total deformation gradient F to mat_user in order to achieve low plasticity, high plasticity, or elastoplastic constitutive model.

The Mat_user.f file is compiled to generate the user-defined UserMat.dll. The AdvantEdge FEM engine uses the *.dll file to calculate the material state. The material parameters are specified in the projectname_wp.twm file, and the AdvantEdge FEM engine outputs the user-defined state variables of the project.tec file. At the top of the projectname_wp.twm file, “MODELTYPE” must be changed to “USER-DEFINED- MATERIAL“ to invoke user-defined material parameters. The spin tensor $W\sigma^{n}+\sigma^{n}W$ is applied outside the subroutine.

The parameter eps (3, 3), representing the deformation gradient, is used to calculate the updated Cauchy stress $\sigma^{n+1}\;$, and the results are returned to the parameter sig(3, 3) representing the Cauchy stress. In the projectname_wp.twm file, there are some reserved variables such as: YOUNG, POISSON, SIGMA0, CONDUCTIVITY, HEATCAP, DENSITY, etc. These variables are defined from UMATPAR01 to UMATPAR50 and assigned to user-defined material parameters. These variables can be passed to the material properties of the user-defined program using the projectname_wp.twm file. The projectname_wp.twm file can be used to transfer these variables to the material properties in the user-defined program, while these variables remain unchanged throughout the execution process. The stress update algorithm flow is shown in Figure 3.

#### 3.2.3. User Material Subroutine Interface and Main Parameters

AdvantEdge FEM provides an interface subroutine called mat_user. When the constitutive model is imported into AdvantEdge FEM, the user subroutine is compiled into a dynamic link library for dynamic loading. A subroutine is used to calculate the stress variation of materials, and Cauchy stress is returned to AdvantEdge FEM. The engine passes both the deformation rate D and the total deformation gradient F to the mat_user, in order to build low-plastic, high-plastic, or elastoplastic constitutive models.

The mat_user.f file is compiled to generate a custom UserMat.dll file. This *.dll file is used to calculate the material state by the AdvantEdgeFEM engine. Material parameters are specified in the projectname _ wp.tm file, and the Advant Edge FEM engine outputs user-defined state variables in the project.tec file. At the top of the projectname_wp.twm file, “MODELTYPE” must be changed to “USER-DEFINED-MATERIAL” to invoke user-defined material parameters. The spin tensor $W\sigma^{n}+\sigma^{n}W$ is applied outside the subroutine.

The updated Cauchy stress $\sigma^{n+1}$ is calculated by the parameter eps (3, 3), representing the deformation gradient, and the results are returned to the parameter sig (3, 3) representing Cauchy stress. Parameter deps (3,3) represents [ deformation ] × [ time increment ], and assigns the Cauchy stress sig (3,3), engine _ s (1: 20) and user _ s (1: 100) to the user. Users can give them new physical definitions where the variable user_s(1:5) can be used to represent the ‘USER’ keyword in the projectname.inp file.

There are at most 100 state variables that can be defined in addition to stress updates, such as plastic strains, hardening parameters, etc. The AdvantEdge engine also has some state variables that need to be updated. Therefore, it is necessary for users to update these retained state variables in subroutines, including engine_s(1) plastic strain, engine_s(3) plastic work rate, engine_s(4) plastic strain rate, engine_s(5) damage.

There are some reserved variables in the projectname_wp.twm file, such as YOUNG, POISSON, SIGMA0, CONDUCTIVITY, HEATCAP, DENSITY, etc. These variables are defined from UMATPAR01 to UMATPAR50 and assigned to user-defined material parameters. These variables can be transmitted to the material properties of the user-defined program with the projectname_wp.twm, file and remain unchanged throughout the execution.

A total of 50 material parameters are allowed to be defined from UMATPAR01 to UMATPAR50. These parameters will be passed to the user subroutine through the D (1: 50) array, and the definitions should be consistent with those in user subroutines.

#### 3.2.4. Establishment and Compilation of User Material Subroutine

Enter the AdvantEdge FEM system and select Custom Materials ► Constitutive Model ► Customized material constitutive, as shown in Figure 4.

(1)Initialization

It is necessary to correctly initialize variables such as material state at the beginning of the subroutine, including the definition of system reserved parameters and user parameters.

(2)The material parameters are transmitted to the system, and the improved constitutive model considering recrystallization softening is calculated. The radial regression method is used to calculate the stress update.(3)Compilation

Compile the subroutine, as shown in Figure 5.

## 4. Finite Element Simulation of High-Speed Milling Titanium Alloy Ti-6Al-4V

The properties of Ti-6Al-4V are shown in Table 4, including thermal conductivity, specific heat capacity, and thermal expansion coefficient. The workpiece is divided into 90,000 units; tools are rigid bodies, $r_{\varepsilon }=0.8$ mm, $\gamma_{0}=5{}^{\circ}$, which divide into 180,000 units. The mesh of the tool tip and cutting zone is the finest, and the minimum unit size of the workpiece and tool are set to 0.08 mm and 0.024 mm, respectively. The workpiece size is set to 10 mm × 2 mm. The tool length is set as 1 mm in order to reduce the computation time. Setting thermal boundary condition allows heat conduction between the tool and the workpiece, allowing for a rapid temperature rise in the tool.

A set of data extracted from the stress–strain curve (Figure 1 and Figure 2) is shown in Table 5. The finite element simulation was carried out according to the milling test (Table 3) and is shown in Table 6. The stress is approximately 460 MPa at 950 °C, and it reduced to 414 MPa (J–C constitutive model) and 220 MPa (improved model), respectively When the temperature rises to 1000 °C, the stress of the J–C constitutive does not change significantly when the temperature rises above the recrystallization point; however, the improved model has a significant decline.

It can be seen from the above analysis that the improved model is closer to the data obtained from the SHPB test, which proves that the stress would drop as the recrystallization softening effect, while the temperature of Ti-6Al-4V reaches the recrystallization point. The improved model is more suitable for practical applications in high-speed cutting production, and provides important guidance for parameter selection, temperature control, and tool selection in production.

## 5. Conclusions

(1)An improved J–C constitutive model containing two expressions is established by the SHPB test and milling test, such that the softening effect of recrystallization is considered. This can describe the stress–strain change trend of high-speed cutting Ti-6Al-4V. There is no dynamic recrystallization phenomenon in the early stage of deformation, as the deformation amount and the temperature are not high enough, and the J–C constitutive can well reflect the deformation process of the material at this point. When the cutting temperature gradually grows to the recrystallization point and the strain is less than the critical value, the ratio of recrystallization changes is added to the constitutive by the integer function, which reflects the softening effect of dynamic recrystallization. The greater the deformation, the higher the strain. Until it reaches the critical strain value, the constitutive shows the stress softening process, and the softening degree is reflected by the parameters *s* and *t* in the constitutive. The improved model can better reflect the stress–strain relationship at high temperature and heavy pressure, while the recrystallization softening effect is not reflected in the previous constitutive model studies.(2)The relevant theories of cutting finite element simulation have been studied, such as nonlinear constitutive elastic–plastic deformation, strain state, and material yield. A subroutine, including the Recht shear failure instability criterion and the improved model, is coded in Fortran and embedded in the finite element simulation software AdvantEdge FEM with the return mapping stress integration algorithm.(3)It can be seen from the finite element simulation between the J–C constitutive model and the improved one that the simulated stress of the improved model drop dramatically from 460 MPa to 220 MPa when the temperature rises from 950 °C to 1000 °C, and its decline reaches 46.7%, while the J–C model only decreased by 10%. Comparative studies indicate that the stress change of the improved constitutive simulation is closer to the SHPB test results than the J–C constitutive, and the new one is more suitable when it expresses the high temperature and heavy impact in high-speed milling.

## Figures and Tables

**Figure 1 materials-16-03344-f001:**
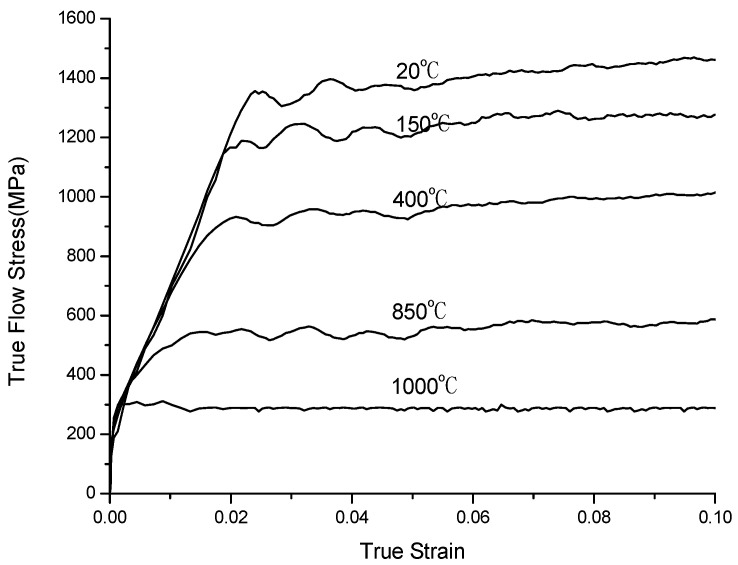
Stress–strain relationship at strain-rate 2000 s^−1^.

**Figure 2 materials-16-03344-f002:**
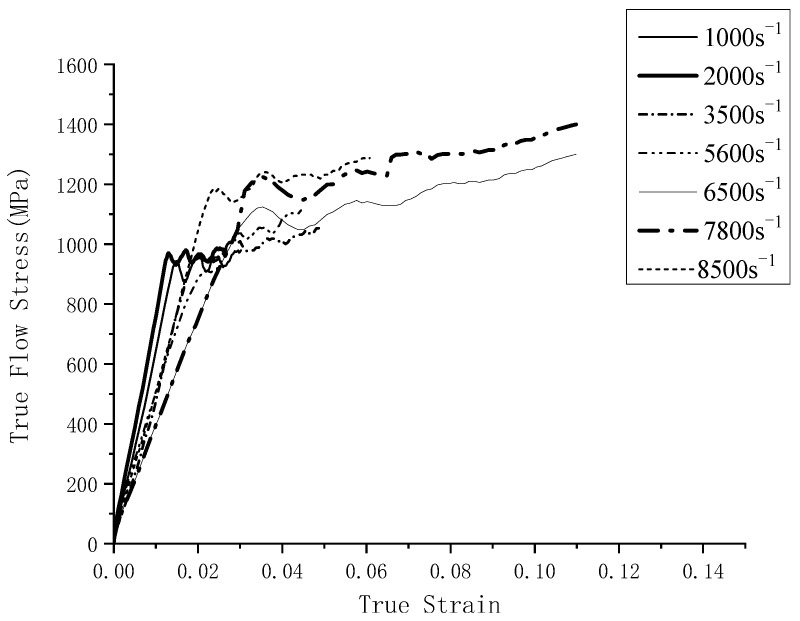
Stress–strain relationship of different strain rates at room temperature.

**Figure 3 materials-16-03344-f003:**
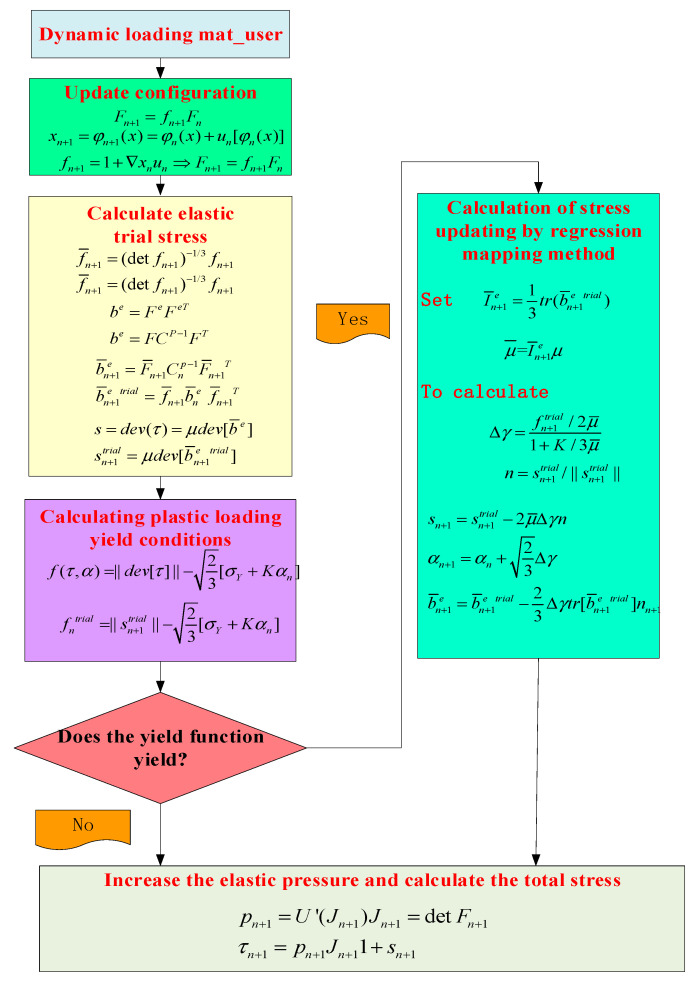
Subroutine flow chart.

**Figure 4 materials-16-03344-f004:**
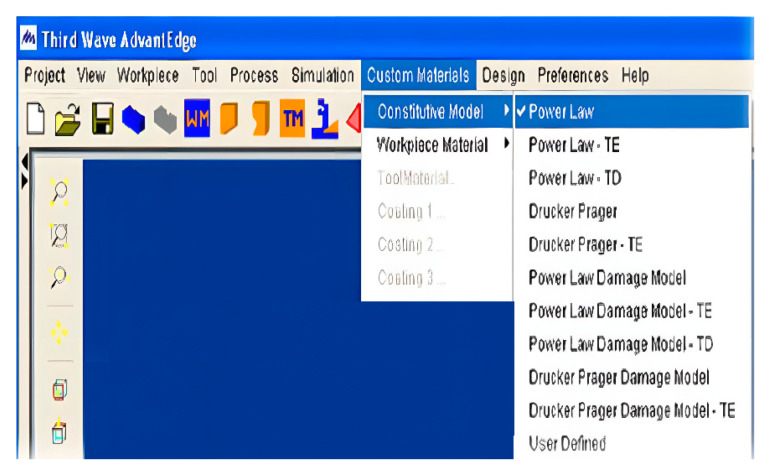
User constitutive selection menu.

**Figure 5 materials-16-03344-f005:**
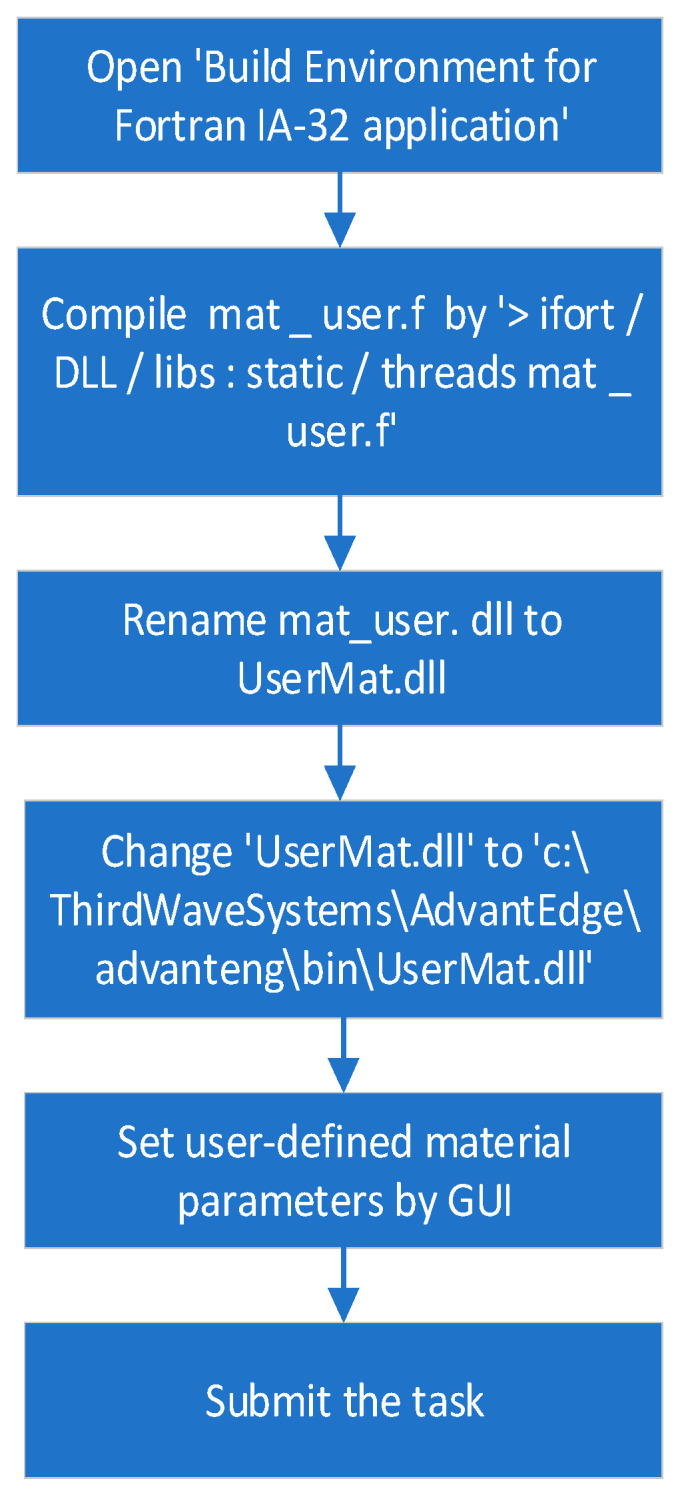
Compile flow chart.

**Table 1 materials-16-03344-t001:** Ti6Al4V composition table (%).

C	Si	Fe	Ti	Al	N	V	S	O	H	Y
0.11	<0.03	0.18	Bal.	6.1	0.007	4.0	<0.003	0.11	0.0031	<0.005

**Table 2 materials-16-03344-t002:** Physical properties of Ti6Al4V.

Density	Elastic Modulus	Strength Limit	Melting Point	Thermal Conductivity	Elongation	Yield Limit	Specific Heat Capacity
$\mathrm{kg}/m^{3}$	(GPa)	MPa	(°C)	(w/m·°C)	(%)	(MPa)	$C(\frac{J}{\mathrm{kg}\cdot {}^{\circ}C})$
430	113.8	950	1668	7.3	14.0	820	526

**Table 3 materials-16-03344-t003:** The test scheme and the equipment.

Test	Equipment		Scheme	
SHPB test	The diameter of the pressure bar: 14 mm Striker bar length: 200 mm Length of incident rod: 400 mm Transmission rod length: 400 mm	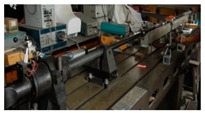	Specimen size (mm): *ϕ*8 × 4, *ϕ*7 × 3.5, *ϕ*6 × 3 Strain rate (s^−1^): 1000~8500 Temperature (°C): 20~1000	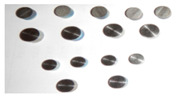
		SHPB experimental apparatus		Specimen
High speed milling	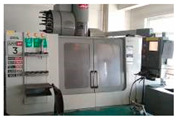	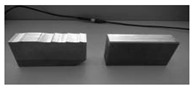	Milling speed *v_c_*/(m/min): 40, 60, 80, 100, 120 Feed per tooth *f_z_*/(mm/r): 0.05, 0.08, 0.11, 0.14, 0.17 Cutting depth *a_c_*/(mm): 0.4, 0.7, 1.0, 1.3, 1.6 Cutting width *a_e_*/(mm): 1.0, 1.5, 2.0, 2.5, 3.0	
	Haas Machining Center VF-3	Specimen		

**Table 4 materials-16-03344-t004:** The material properties.

Material Properties	Value
Thermal conductivity (W/m·K)	$e^{0.0011*T}$
Specific heat capacity (J/gK)	2.24$e^{0.0007*T}$
Thermal expansion coefficient (m/K)	9.4$e^{-006}K^{-1}$

**Table 5 materials-16-03344-t005:** Comparison of the flow stress corresponding to different temperatures.

Temperature (°C)	150	600	850	900	950	1000	1100
Flow stress (MPa)	1186.6	708.4	593.2	555.3	437.6	298.3	218.6

**Table 6 materials-16-03344-t006:** Finite element simulations.

Simulation Conditions	The Mises stress simulation under the condition of a temperature of 950 °C	The Mises stress simulation-based J–C constitutive model under the condition of a temperature of 1000 °C	The Mises stress simulation-based improved model under the condition of a temperature of 1000 °C
Finite Element Simulations	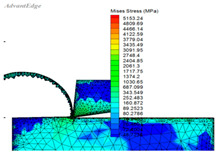	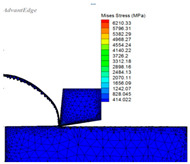	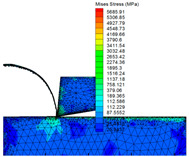

## Data Availability

All relevant data are within the paper.

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
