# Peer review of "Subroutine Embedding and Finite Element Simulation of the Improved Constitutive Equation for Ti6Al4V during High-Speed Machining"

_materials, 2023, doi:10.3390/ma16093344_

Round 1

Reviewer 1 Report

materials-2273026-peer-review-v1

The general topic id of interest, however there are serious flaws in the scientific support.

The Authors start with a JC equation supposedly adding a recrystallization. However the used  equations 2 and 3 are not supported on analytical or experimental evidence.

Later on the Use of Recht is not necessarily compatible with previous equations.

Recht can be a possible validation of th JC modification, bot embedding it with eq 2 and 3 which are initially not proven adds to the inconsistency.

The text is not well written, there are strange fonts changes and Chinese characters.

The work is on Ti6LA4V however table 1 is for another alloy. the work seem random without any scientific contribution.

Reviewer 2 Report

I want to congratulate the authors for the manuscript titled as “Subroutine Embedding and Finite Element Simulation of the Improved Constitutive Equation for Ti6Al4V During High Speed Machining”. The article is interesting and deserves the attention of readers. However, there are several points in the article that require further explanation.

 First, the cutting material is widely used and preferred in the industrial applications and literature papers which make it difficult to address a novel approach. Please clearly define the new sides of this paper in abstract and required points respectively.

The authors should add more information about the experimental method and the test equipment specification.

Abstract: writing is too generalized, and it is too long especially for explanation about the material and method process. The main theme of this paper is not described in the abstract. Abstract section should be concisely reflected the content and summarize the problem, the method, the results, and the conclusions. The abstract needs to be improved. Demonstrate in the abstract novelty, practical significance. The author should add a sentence of research background to the abstract. Please add more qualitive and quantitative results of your work.

Each one of the cited references  must be discussed individually and demonstrate their significance to your work. Not [1-3], should be [1] text what is presented in the manuscript [1] text what is presented in the manuscript [2].

After analyzing the literature, show before formulating the goal of the "blank" spots. Which has not been previously done by other researchers? You must show the importance of the research being undertaken. Show what will be the new research approach in this article. You need to show a hypothesis. In the last paragraph of the introduction, add scientific novelty and practical relevance. Add a clear purpose to the article.

The authors must give the actual photos belong to experimental infrastructure. Also, a graphical abstract would be helpful for the presentation of the study.

Improve the results and discussion and conclusion parts. The results and discussion section should be widened with more focusing point of the findings. And these sentences should be supported with the literature studies. Results and discussion and conclusion parts are inadequate according to citation and analyze in detail. There should be the importance of the study in detail, comparison results with other approaches in literature, the success of the prediction and computational results.

The conclusions section needs to improve with selected and highlighted main findings. In conclusion section, it is necessary to more clearly show the novelty of the article and the advantages of the proposed method. Add qualitative and quantitative results of your work. Please try to emphasize your novelty, put some quantifications, and comment on the limitations. This is a very common way to write conclusions for a learned academic journal. The conclusions should highlight the novelty and advance in understanding presented in the work.

Language used in the manuscript is generally satisfying. However, writers should pay more attention of singular / plural nouns. Also, they should control the spell check/ punctuation of words and sentences. Please check all manuscript for language and misspellings. Also, please recheck upper and lower case letter. In addition, spaces should be added between words and numbers. The authors can use suitable grammar-checking software / use the help of a native English speaker to correct these mistakes. Please fix the typographical and eventual language problems in paper.

References are not enough. Such a work deserves many citations. Minimum 10-15 references need to be added and some of them should be discussed. (Please only cite most relevant articles)

Reviewer 3 Report

The authors have implemented a method combining theoretical analysis with experimental and finite element simulation in the high-speed machining of Ti-6Al-4V alloys. Overall, the authors have tried well to simulate the milling process using FEM software.  

The following corrections are required:

1. In the Abstract, Mention the details of the experimental part (details of the cutting tool), Define high-speed machining in starting of the abstract. Also the sentence "While the constitutive models study is the key technology in the high-speed machining mechanism research" is not appropriate, Rephrase it. Also, Mention the results benefits in % by using Modified Model in simulation. Comparative results must be addressed.  

2. There are many research reported the use of modified J-C Models in Ti-6Al-4V machining, In which aspects your model is different, mentioned in the last part of the Introduction?

3. Introduction is shallow, only 17 references were cited, which ensured that the literature review was not addressed properly. Add more relevant references and write the literature gap in the last paragraph of the Introduction. 

4. Add the references for Table 1 and Table 2. Also, Put the required citations for all Equations mentioned in the study.

5. Figure 1 quality is very poor, Replace it with good quality flow chart.

6. Figure 2  is very poor, Replace it with a good-quality picture.

7. Write the English language in Figure 3, Also improve the image quality. 

8. Why you have chosen a very small workpiece size (10mm×2mm). This is not realistic.

9. Milling speed must be replaced by cutting speed in the entire manuscript. Also, as the maximum speed is 100 m/min, then how you can say that the machining is high speed? 

10. In section 4, the Experimental part is inadequate. Give the details of machining length, cutting tool name, geometry, and coatings. Also, What responses were compared in the simulation? I saw only chip morphology is compared which is the least important index for machining. Based on this analysis, it can't say that the modified Model is giving better results than the J-C model.

I suggest to compared tool wear and surface roughness experimental results with Simulation. 

11. Conclusion part needs to be improved by adding a future scope. Also, write major findings only.

Major Resion Required

Round 2

Reviewer 1 Report

There’s a serious scientific flaw which the authors didn’t consider :

Use of Recht is not necessarily compatible with previous equations.Recht can be a possible validation of th JC modification, but embedding it with eq 2 and 3 which are initially not proven adds to the inconsistency.

Reviewer 2 Report

I think the authors have  thorughly improved the paper. It can be accepted in current form

Reviewer 3 Report

The authors have answered all the questions and significant corrections have been made.

Only, English editing is required.

Now in this form, the paper can be accepted.